# Research on Adversarial Domain Adaptation Method and Its Application in Power Load Forecasting

**Min Huang \* and Jinghan Yin**

Department of Software Engineering, South China University of Technology (SCUT), Guangzhou 510006, China
\* Correspondence: minh@scut.edu.cn

**Abstract:** Domain adaptation has been used to transfer the knowledge from the source domain to the target domain where training data is insufficient in the target domain; thus, it can overcome the data shortage problem of power load forecasting effectively. Inspired by Generative Adversarial Networks (GANs), adversarial domain adaptation transfers knowledge in adversarial learning. Existing adversarial domain adaptation faces the problems of adversarial disequilibrium and a lack of transferability quantification, which will eventually decrease the prediction accuracy. To address this issue, a novel adversarial domain adaptation method is proposed. Firstly, by analyzing the causes of the adversarial disequilibrium, an initial state fusion strategy is proposed to improve the reliability of the domain discriminator, thus maintaining the adversarial equilibrium. Secondly, domain similarity is calculated to quantify the transferability of source domain samples based on information entropy; through weighting in the process of domain alignment, the knowledge is transferred selectively and the negative transfer is suppressed. Finally, the Building Data Genome Project 2 (BDGP2) dataset is used to validate the proposed method. The experimental results demonstrate that the proposed method can alleviate the problem of adversarial disequilibrium and reasonably quantify the transferability to improve the accuracy of power load forecasting.

**Keywords:** domain adaptation; adversarial learning; adversarial equilibrium; transferability quantification; power load forecasting

**MSC:** 68T07

## 1. Introduction

Power load forecasting aims to predict the power load in the power system in the future by mining the characteristics of users' power consumption behavior hidden in historical records, weather, dates, and other data. According to the forecast time, power load forecasting can be divided into long-term, medium-term, and short-term. Short-term power load forecasting refers to prediction of the power load value several hours or days in the future, which is an important basis for realizing the rapid response of the power system to changes in power load.

Recently, machine learning has accomplished extraordinary triumphs in the avenue of computer vision [1], semantic segmentation [2], regression prediction [3], natural language processing [4], etc. However, two problems of traditional machine learning are gradually exposed: Firstly, traditional machine learning requires a large amount of labeled data, and the cost of collecting and labeling data is expensive; thus, it is difficult to be applied in fields that lack the data required for training models. Secondly, an important condition for traditional machine learning being effective is that test and train data obey the assumption of independent and identical distributions (IIDs); however, the condition of IID is usually not satisfied in the real world, resulting in a decrease in the accuracy and generalization capabilities. Correspondingly, due to the strong personalization of power consumption behavior, there are differences in the distribution of power load data of different users. Due

to the difficulty in collecting historical data, there is a lack of labeled data for training. The above factors hinder the application of traditional machine learning methods in short-term power load forecasting.

Domain adaptation has received extensive attention as one of the effective methods to overcome the difficulties of few-shot learning [5–7]. Domain adaptation aims to transfer knowledge from related labeled data by reducing the distribution difference between the source domain and the target domain. Domain adaptation reduces the number of labeled samples required to achieve the target task and does not strictly require the data to satisfy the condition of IID.

The key aim of the domain adaptation method is to align the feature distribution of the source domain and target domain data. The process of aligning the feature distribution is also called domain alignment. Domain adaptation methods can be divided into three types roughly according to different alignment strategies: discrepancy-based, adversarial-based, and reconstruction-based.

Discrepancy-based methods use different metric schemas to measure the distance between the source domain and the target domain; it aligns the distribution by reducing the difference metric schemas. The method adds different distance loss functions to the artificial neural network. The most widely used metric schemas include Maximum Mean Discrepancy (MMD) [8–10], KL (Kullback–Leibler) divergence [11], JS (Jensen–Shannon) divergence [12], Wasserstein distance [13–15], CORAL (CORrelation ALignment) [16,17], etc.

Adversarial-based methods [18–25] are inspired by GANs and use artificial neural network modules instead of metric schemas to measure the distance. The key components of the adversarial domain adaptation model include a feature extractor and a domain discriminator. The feature extractor extracts the domain-invariant features of the source and target domains to confuse the domain discriminator; at the same time, the domain discriminator distinguishes a sample from the source domain or the target domain, and the strategy of maximizing and minimizing the domain discrimination loss is used to form a confrontation between the two and to implement domain alignment during the adversarial training.

Reconstruction-based methods [26–29] aim to reconstruct all domain data under the premise of preserving domain-specific features to better help learn domain-invariant features. The encoder–decoder is a typical implementation of reconstruction-based methods, the shared encoder encodes the input data as hidden features and learns domain-invariant features, and the decoder reconstructs the hidden features and preserves domain-specific features.

Domain adaptation methods realize the cross-domain transfer and reuse of knowledge, and so many researchers use it to overcome the problem of data shortage in power load forecasting: Ref. [30] proposes a general framework for adversarial domain adaptation methods on time series prediction problems; Ref. [31] introduces a contrastive evaluation module to protect the task-specific features of the target domain in domain alignment; Ref. [32] builds adversarial feature capture networks to achieve reliable energy prediction. Ref. [33] proposes an electricity load forecasting algorithm through bidirectional generative adversarial networks and validates it on user data with different behavior patterns; the flexibility and accuracy of the algorithm are improved. Ref. [34] proposes to construct a time-independent model by maximizing the segmentation of time series differences to suppress the unstable prediction accuracy caused by the time distribution shift. The above studies focus on solving the problem that traditional machine learning relies on a large amount of labeled data and cannot learn knowledge from non-IID data. However, the methods do not consider the problem of lack of transferability quantification, and the adversarial-based methods [30,33] do not consider the problem of adversarial disequilibrium. Both of the above two problems will lead to the decline of the accuracy of the domain adaptation method and the robustness of the model. Therefore, this paper focuses on analyzing and researching these two problems and their solutions.

The main contributions of this paper include:

- This paper proposes a novel adversarial domain adaptation method, which alleviates the adversarial disequilibrium problem through the initial state fusion strategy and quantifies transferability by calculating domain similarity based on information entropy.
- The proposed method is used for power load forecasting, which improves the accuracy of power load forecasting with a small amount of data.
- This paper compares and analyzes the proposed method with a variety of baselines. The results show that the proposed method can effectively maintain the adversarial equilibrium and reasonably quantify the transferability.

The rest of this paper is organized as follows: Section 2 analyzes two problems and summarizes the current solutions; Section 3 details the framework of the proposed method; Section 4 shows the experimental content and the analysis of the results; Section 5 concludes this article.

## 2. Related Work

This section briefly summarizes the current solutions for the adversarial disequilibrium and the approaches to design metrics of transferability.

### 2.1. Adversarial Disequilibrium Problem

For adversarial-based methods, the domain discriminator distinguishes whether they originate from the source domain or the target domain according to the features generated by the feature extractor; the domain discriminant results make a key impact on the parameter update of the model. However, the feature extractor easily wins the competition when it only retains shallow feature representation and discards the deep feature representation, which leads to the fact that the domain discriminator cannot accurately reflect the distance in distribution. The methods for solving the adversarial disequilibrium problem can be divided into two categories according to different enhancement strategies.

One way to address this problem is to combine the different metrics, which means the metric is introduced in adversarial training, and the training goal is to confuse the discriminator and reduce the metric. When adversarial disequilibrium occurs and the domain discriminator fails, the model can continue to optimize parameters according to the metric, so the method can effectively improve the training stability. Difference metrics have been maturely applied, but they are suitable for different scenarios due to differences in measurement dimensions, time overhead, gradient information, etc. Therefore, an effective selection from numerous metrics becomes the key to the feasibility of the method. Ref. [35] adopts Maximum Density Divergence (MDD) to minimize inter-domain distance and maximize intra-domain density, and embeds MDD into an adversarial-based domain adaptation framework to overcome the adversarial disequilibrium problem. Ref. [36] combines Multi-Kernel Maximum Mean Discrepancy (MK-MMD) reduces the fluctuation of the training process and maintains the adversarial equilibrium; Ref. [37] integrates MK-MMD in the partial adversarial domain adaptive network to deal with the adversarial disequilibrium problem.

Domain discriminator augmentation increases the domain information contained in the input features of the domain discriminator. From the view of the adversarial game, the method adds information to the domain discriminator for avoiding it being in a weak position in the confrontation. The stronger the domain discriminator, the better it can guide the feature extractor to learn domain-invariant features in adversarial. Ref. [38] proposes a conditional adversarial domain adaptation method, which supplements category information in the input features of the domain discriminator, and uses a multi-linear mapping method to describe the joint representation of feature information and category information. Ref. [39] combines features and labels to help model learning discriminative features, and proposed the principle of entropy minimization to set reliable pseudo-labels for the target domain. Ref. [40] proposed to normalize the conditional information so that it has the same norm as the feature, expand the conditional output norm, and improve the conditional

strategy based on the prototype. Ref. [41] proposes that the sample adversarial domain adaptively converts the noncentral sample distribution to the central sample distribution to improve the classification degree of feature distribution, and indirectly adds category information to the input of the feature extractor through clustering methods.

### 2.2. Lack of Transferability Quantification Problem

Domain adaptation learns domain-invariant features by reducing the distribution distance between the source domain and the target domain and then transferring knowledge from the source domain to the target domain. However, not all source domain knowledge can promote the achievement of the target task. Traditional domain adaptation methods lack the contribution differentiation of source domain knowledge. Useless information and noise in the source domain will hinder the model from achieving the target task, which will eventually lead to the degradation of method performance and the occurrence of negative transfer. The similarity-based quantification of transferability is currently an effective method for alleviating this problem.

The similarity-based transferability quantification method is based on the assumption that the higher the similarity is, the higher the transferability is, and the contribution of the source domain to the target task is distinguished according to the domain similarity, and the knowledge that is conducive to achieving the target task is selectively transferred. The key to this method is how to quantify domain similarity. Ref. [42] proposes an attention mechanism to quantify domain similarity, enhance semantic information with high transferability between domains and within domains, and improve the generalization ability and robustness of the algorithm. Ref. [43] proposes a weighted moment distance to quantify domain similarity, enhance the impact of high domain similarity data on the transfer process. Ref. [44] fuses batch spectral penalty in an adversarial-based domain adaptive network to suppress the phenomenon of forced alignment of low-transfer features, and enhance method transferability and discriminating ability.

## 3. Proposed Method

This section mainly introduces the novel method: Section 3.1 proposes an initial state fusion strategy to maintain the adversarial equilibrium, Section 3.2 designs a selective transfer method based on information entropy, and Section 3.3 details the architecture of models.

### 3.1. Adversarial Equilibrium Strategy Based on Initial State Fusion

The key of the domain discriminator augmentation is to supply domain structure information to the features, thereby improving the reliability of the domain discrimination and avoiding adversarial disequilibrium; therefore, the information introduced in the features has a crucial impact on the effectiveness of the method.

The initial state refers to the original data without feature extraction and distribution alignment, which has the most complete domain structure information, and the statistical features of the source domain and target domain data are highly distinguishable. These characteristics meet the requirements of the information for implementing domain discriminator augmentation. Therefore, this paper proposes to fuse the initial state in the input features of the domain discriminator. The reliability of the domain discrimination results is improved by supplementing the domain structure information of the input features. It avoids the domain discriminator being weak in the adversarial training and finally realizes the domain discriminator to reflect the distance of distribution implicitly and more accurately.

Due to the large dimensional difference between the intermediate features and the initial state, conventional feature fusion operations such as concat and add are easy to fail. We propose a strategy of splitting features first and then fusing them. Critical steps are shown in Figure 1. Firstly, the domain features (yellow in Figure 1) of the data are extracted using the feature extractor. Secondly, the domain features are split into several subfeatures

with dimensions equivalent to the initial state (pink in Figure 1), and subfeatures gradually dot the product with the initial state; the dot product is given by

$$a \bullet b = \sum_{i=1}^{n} a_i b_i = a_1 b_1 + a_2 b_2 + \cdots + a_n b_n \tag{1}$$

where $a$ and $b$ represent the subfeature and the initial state, respectively, and $a_i$ and $b_i$ represent the $i$-th element.

Each subfeature will perform the operation of (1) with the initial state; new feature elements are merged to form the fused feature (red in Figure 1). Finally, the fused feature is input into the domain discriminator for domain discrimination.

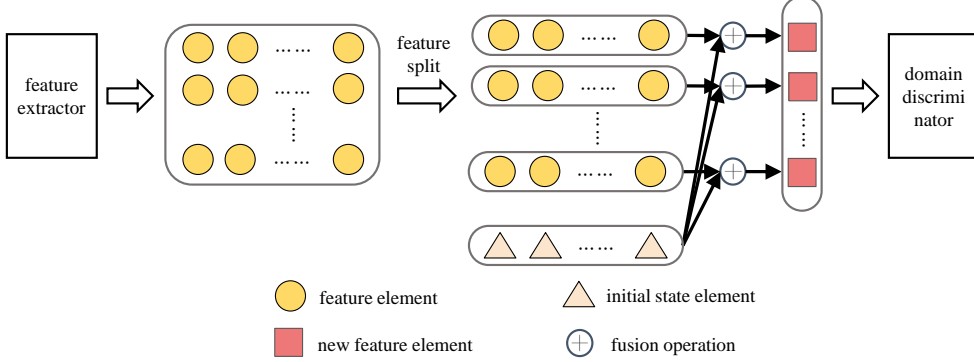

**Figure 1.** Initial state fusion strategy.

### *3.2. Transferability Quantification Based on Information Entropy*

The quantification of transferability is based on the premise that domain similarity and transferability are positively correlated. In the adversarial domain adaptation method, the information entropy of domain discrimination can objectively reflect domain similarity. Therefore, we propose a transferability quantification method based on information entropy, which realizes the transfer source domain samples selectively and inhibits the occurrence of negative migration to a certain extent.

In information theory, information entropy is used to measure the information content of an event. The smaller the probability of an event, the greater the amount of information it contains, and the information entropy also increases. $p(x_i)$ is used to represent the probability density of event $x_i \in X$, $i = 1, 2, \ldots, n$, and the information entropy of event $X$ is calculated by

$$H(X) = -\sum_{x_i \in X} p(x_i) ln p(x_i) \tag{2}$$

The domain discrimination is the basis for the adversarial domain adaptation method to reflect the degree of feature distribution alignment. The essence of domain discrimination is a two-class prediction task of the sample belonging to the source domain or the target domain. When the output layer of the domain discriminator is activated by the Softmax function, the output after activation is two predicted values whose sum equals 1, denoted as $[p_s, p_t]$, which respectively represent the probability that the domain discriminator thinks the sample belongs to the source domain or the target domain. The Softmax activation is calculated by

$$S_i = \frac{e^i}{\sum_{j=1}^{n} e^j} \tag{3}$$

The information entropy of the domain prediction value is used to reflect the domain similarity. The closer the outputs $p_s$ and $p_t$ of the domain discriminator are, the more successfully the features of the source domain sample confuse the domain discriminator, making it impossible to make accurate domain discrimination. Furthermore, the high domain similarity means that the information entropy of the domain prediction value is

maximized, and the source domain samples that generate this feature should be given a higher weight during the transfer process. The weight is calculated by

$$\omega_i = exp[-p_s ln(p_s) - p_t ln(p_t)] - 1 \qquad (4)$$

where the exponential is the information entropy of $p_s$ and $p_t$.

We propose to quantify transferability based on information entropy to tackle the problem of the lack of transferability quantification method, by weighting the source domain samples according to the quantification results to transfer knowledge selectively. The process of transferability quantification is shown in Figure 2. Firstly, the features of samples are extracted. Samples with high domain similarity are shown as having more domain-invariant features in the feature space, and the feature distribution of the source domain and target domain has a high degree of coincidence. Then, make the domain discrimination; the smaller the difference between the $p_s$ and $p_t$ output by the domain discriminator, the higher the similarity that the samples have, and the richer the transferable knowledge that is contained. At this time, the information entropy of the domain discrimination increases. Finally, calculate the weights; samples with higher transferability cause a greater impact on the transfer.

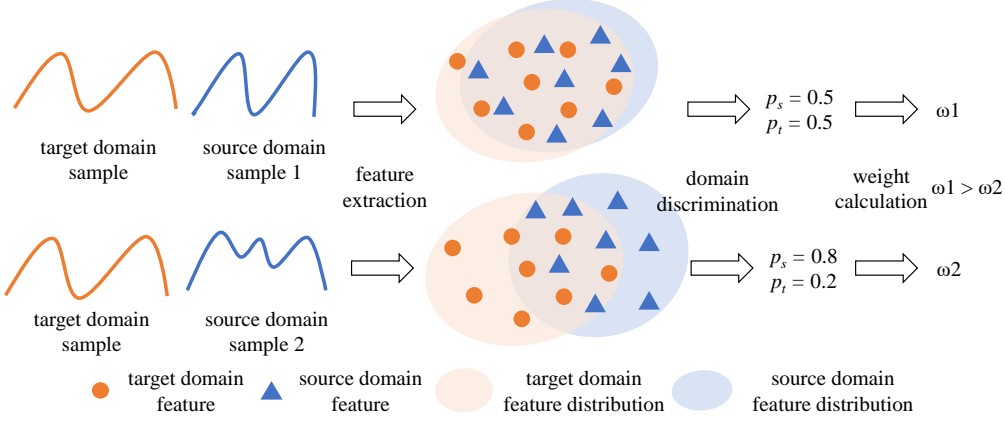

**Figure 2.** Transferability quantification process.

### 3.3. A Novel Adversarial Domain Adaptation Method

3.3.1. Model Structure

The one-dimensional convolutional neural network and Bidirectional Long Short Term Memory Networks (1DCNN-BiLSTM) has both the efficient feature extraction ability of 1DCNN and the advantages of BiLSTM in describing the dependencies of a time series [45,46]. We use 1DCNN to build a feature extractor and BiLSTM to build a predictor; the model structure is shown in Figure 3. The model consists of three basic modules, a feature extractor, predictor, and domain discriminator. In addition, the initial state fusion module (the light blue module in Figure 3) is added before the domain discriminator, and the transferability quantification module (light green module in Figure 3) is added after the domain discriminator.

The model hyperparameters are shown in Table 1. The column hyperparameter are the properties required to build the model, followed by the corresponding values. The first line indicates that the feature extractor has three layers of 1DCNN. The values in the brackets in the second row represent the respective kernel size of the aforementioned three layers. The source domain and target domain data are convolved with 1DCNN to generate domain-invariant features. Dropout [47] is used in the BiLSTM layer of the predictor to randomly suppress neurons to avoid model overfitting. The features are fused with the initial state, and domain discriminant results are used to calculate the total loss.

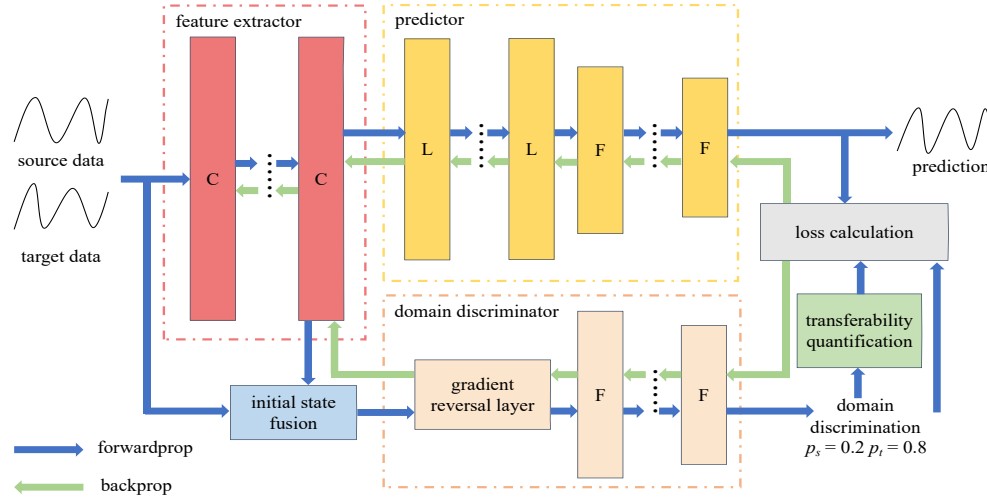

**Figure 3.** Model structure. C represents 1DCNN, L represents BiLSTM, F represents fully connected layer.

**Table 1.** Model Hyperparameters.

| Module | Hyperparameter | Value |
|---|---|---|
| feature extractor | Layer of 1DCNN | 3 |
| | Size of the convolving kernel by each layer of convolution | (3, 3, 3) |
| | The number of channels produced by each layer of convolution | (64, 64, 64) |
| domain discriminator | Layer of Dense | 2 |
| | Size of each output sample by each layer of Dense | (32, 2) |
| predictor | Layer of BiLSTM | 2 |
| | The number of features in the hidden state by each layer of BiLSTM | (64, 64) |
| | Dropout probability | 0.5 |
| | Layer of Dense | 2 |
| | Size of each output sample by each layer of Dense | (32, 1) |

The domain discriminant loss is composed of the cross-entropy between the domain discriminantion and the real domain label, which is calculated by

$$Loss_{dcls} = \frac{1}{n_s} \sum_{i=1}^{n_s} L_{ce}(d_s^i, y_s^{di}) + \frac{1}{n_t} \sum_{i=1}^{n_t} L_{ce}(d_t^i, y_t^{dt}) \tag{5}$$

The prediction loss consists of two parts: the weighted source domain prediction loss and the target domain prediction loss, which is calculated by

$$Loss_{pred} = \frac{1}{n_s} \sum_{i=1}^{n_s} \omega_i (y_s^i - y_s^{pi})^2 + \frac{1}{n_t} \sum_{i=1}^{n_t} (y_t^i - y_t^{pi})^2 \tag{6}$$

The total loss of the model is composed of the domain discrimination loss and the prediction loss, which is calculated by

$$Loss = Loss_{dcls} + Loss_{pred} \tag{7}$$

where subscript *s* indicates that the variable belongs to the source domain, subscript *t* indicates that the variable belongs to the target domain, *n* is the number of samples in the

domain; $d^i$ is the domain label, $y^{di}$ is the predicted domain label, $y^i$ is the true value, $y^{pi}$ is the prediction, $\omega_i$ is the weight, and $L_{ce}$ is the cross-entropy loss function.

### 3.3.2. The Critical Steps of the Algorithm

The algorithm flow is shown in Figure 4. The critical steps of each epoch during training include:

1. Feature extraction; the feature extractor performs feature extraction and distribution alignment on the source domain and target domain data to generate domain-invariant features.
2. Initial state fusion; the domain-invariant features are split into sub-features, and the sub-features are gradually fused with the initial state to generate fused features.
3. Prediction and domain discrimination; the input domain-invariant features into the predictor and output predicted values, and input the fused features into the domain discriminator and output domain discriminant values.
4. Transferability quantification; measure the domain similarity according to the domain discriminant value and calculate the weight of the source domain samples.
5. Loss calculation; calculate the prediction loss and the domain discrimination loss separately, then obtain the total loss.
6. Model parameter optimization; the gradient information is calculated based on the loss value, and the model parameters are updated through the preset optimizer.

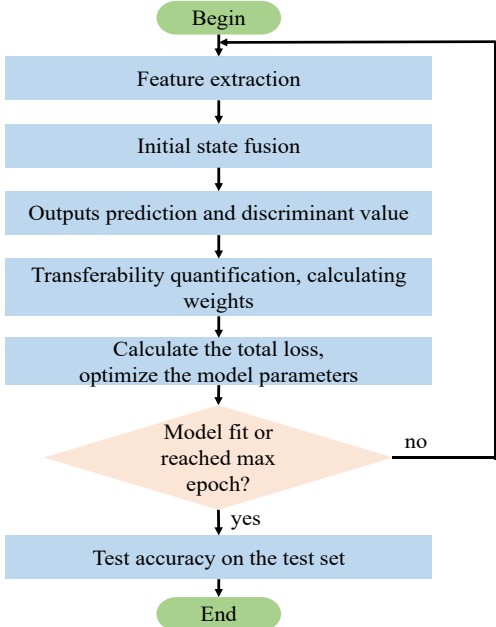

**Figure 4.** Algorithm flow chart.

## 4. Experimental Setup and Results

In this section, we extensively evaluate our approach and compare it with state-of-the-art domain adaptation methods. We also provide a detailed analysis of the proposed framework, demonstrating empirically the effect of our contributions.

### 4.1. Datasets

We evaluate the proposed approach to the BDGP2 dataset [48]. The time range is from 2016 to 2017. The sampling interval is 1 h. The sampling value includes power load, heating, cooling water, steam, and other meter data; in addition, this data set integrates outdoor temperature, humidity, cloud cover, and other climatic factors that can affect power consumption.

Four residential buildings are selected for analysis, namely Bear_lodging_Evan (domain A), Robin_lodging_Renea (domain B), Rat_lodging_Ardell (domain C), and Fox_lodging_Angla (domain D); the load has a periodic characteristic with the user's living habits, which is shown in Figure 5. We use the Augmented Dickey Fuller (ADF) to test that the time series is stationary. The *p* value is 0.00000218, and the absence of missing values is also the important reason for selecting the mentioned building's data. The variables of the inputs are shown in Table 2.

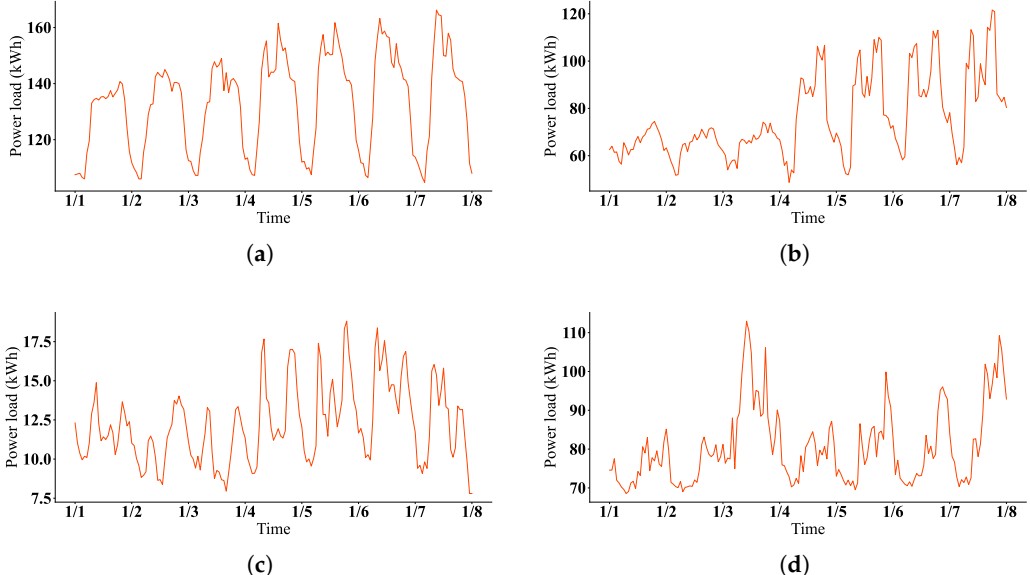

**Figure 5.** Power load for the four buildings. (**a**) Building A; (**b**) building B; (**c**) building C; (**d**) building D.

**Table 2.** The Dataset Variables of Model Inputs.

| Variables | Units | Definition |
| --- | --- | --- |
| TimeStamp | - | Date and time in the local timezone |
| Load | kWh | The sum of the electric power used over a certain time |
| AirTemperature | °C | The temperature of the air in degrees Celsius |
| DewTemperature | °C | The temperature to which a given parcel of air must be cooled at constant pressure and water vapor content for saturation to occur |
| SeaLevelPressure | hPa | The air pressure is relative to the mean sea level |
| WindSpeed | m/s | The rate of horizontal travel of air past a fixed point |

The experiment adopts single-step time series forecasting, the input are the variables in Table 2 of the first 24 h in each sliding window, and the true value is the load of the next hour. To verify the effectiveness and accuracy of the proposed method, we construct 12 transfer tasks for each method, and each task is denoted as S→T, which means the S is the source domain and the T is the target domain. When a building is selected as the source domain, we use all the samples of the building as the source domain data train set. When another building is selected as the target domain, we use 10% of the building's samples as the target domain train set and 20% of the samples as the target domain test set; the remaining 70% of the samples are not used. We use samples from two different buildings to create the condition of non-IID by retaining only a few samples of the target building to simulate the lack of data in the target domain.

### 4.2. Implementation Details

The experiments in this paper are all implemented under the same framework; the programming language is Python3.7.11, the deep learning framework is Pytorch1.10.1, the CUDA11.3, the CUDNN8.2, and the operating system is Windows 10. The CPU is Intel

i5-11400H, the base frequency is 2.7 GHz, the memory is 16 G, the GPU is RTX3050Ti, and the GPU memory is 4 G.

The experiment in this paper adopts the same train setting; the optimizer is Adam, the max epoch is 50, and the batch size is 32, the initial parameters are generated by Pytorch-1.10.1 defaulted, and the learning rate can be calculated as

$$LR = \frac{0.01}{(1 + 10 * p)^{0.75}} \tag{8}$$

where $LR$ is the learning rate of the current epoch, and $p$ is the ratio of the current epoch round to the max epochs.

### 4.3. Results

The objective indicators for the experimental evaluation of prediction accuracy are Root Mean Square Error (RMSE), Mean Absolute Error (MAE), and Mean Absolute Percentage Error (MAPE).

RMSE is sensitive to outliers, and when it is small, it can be considered that the method outputs less predictable values with great deviations. MAE describes the absolute error between the prediction value and the true value, which is the most intuitive. MAPE converts the error value into an error rate, which can evaluate the method performance without considering the order of magnitude of the data.

$$\text{RMSE} = \sqrt{\frac{1}{n} \sum_{i=0}^{n} (y_i - y_{pi})^2} \tag{9}$$

$$\text{MAE} = \frac{1}{n} \sum_{i=0}^{n} |y_i - y_{pi}| \tag{10}$$

$$\text{MAPE} = \frac{100\%}{n} \sum_{i=0}^{n} \left| \frac{y_i - y_{pi}}{y_i} \right| \tag{11}$$

where $n$ is the number of test samples, $y_i$ is the true value, and $y_{pi}$ is the prediction.

The proposed method was compared with FineTune (FT) [49], Wasserstein Distance Guided Representation Learning (WDGRL) [50], Deep Adaptation Networks (DAN) [51], Domain Adversarial Neural Networks–Long Short Term Memory Networks (DANN-LSTM) [52], and Deep CORAL (DCORAL) [53].

FT is the lightest and most widely used method for knowledge transfer. DAN and DCORAL use MMD and CORAL to measure the distance between domains, respectively, which are widely used in discrepancy-based methods. The proposed method, WDGRL, and DANN-LSTM are based on adversarial; however, the difference is our consideration, and attempts to alleviate the adversarial disequilibrium problem. The performances of RMSE, MAE, and MAPE are shown in Tables 3–5. The last row represents the average performance of each method in different tasks, and the best performance of each task is highlighted in bold.

The prediction error of the proposed method is smaller than other methods in most of the adaptation tasks. The proposed method reduces RMSE by 1.53, MAE by 1.29, and MAPE by 1.53%. The reduction in RMSE proves that the method predicts fewer outliers and has a better stability. MAE is used to measure the absolute error, and MAPE is used to measure the error rate. The reduction in the two factors proves that the proposed method can improve the generalization ability of the model and the prediction accuracy effectively.

**Table 3.** RMSE Performance. The best performance of each task is highlighted in bold.

| Task | FT | WDGRL | DAN | DANN-LSTM | DCORAL | Ours |
|------|------|-------|-------|-----------|--------|--------|
| B→A | 25.13 | 15.55 | 13.71 | 13.68 | 14.55 | **11.73** |
| C→A | 27.84 | 14.83 | 15.30 | 13.69 | 16.57 | **12.64** |
| D→A | 22.98 | 13.40 | 17.78 | 13.27 | 15.47 | **12.65** |
| A→B | 13.04 | 10.69 | 12.49 | 11.52 | 10.19 | **9.41** |
| C→B | 14.45 | 9.43 | 11.30 | 9.02 | 9.27 | **8.32** |
| D→B | 16.34 | 11.84 | 13.65 | 14.89 | 10.14 | **9.30** |
| A→C | 3.13 | 2.96 | 2.68 | 2.54 | 2.53 | **2.45** |
| B→C | 3.31 | 4.07 | 3.10 | **2.75** | 3.74 | 2.90 |
| D→C | 2.64 | 3.46 | 2.66 | 2.44 | 2.24 | **2.17** |
| A→D | 14.93 | 13.73 | 10.96 | 11.86 | 11.24 | **10.91** |
| B→D | 18.47 | 16.09 | 11.09 | 11.31 | **9.66** | 10.19 |
| C→D | 17.43 | 13.96 | 13.85 | 13.98 | 10.69 | **9.41** |
| Average | 14.97 | 10.83 | 10.71 | 10.08 | 9.69 | **8.51** |

**Table 4.** MAE Performance. The best performance of each task is highlighted in bold.

| Task | FT | WDGRL | DAN | DANN-LSTM | DCORAL | Ours |
|------|------|-------|-------|-----------|--------|--------|
| B→A | 21.32 | 12.50 | 10.53 | 10.26 | 11.71 | **9.10** |
| C→A | 23.39 | 11.80 | 11.66 | 10.20 | 13.73 | **9.77** |
| D→A | 18.95 | 10.40 | 14.45 | 10.77 | 13.09 | **9.57** |
| A→B | 10.25 | 7.45 | 9.78 | 9.21 | 7.63 | **6.32** |
| C→B | 11.54 | 6.58 | 8.58 | 6.29 | 6.59 | **5.57** |
| D→B | 13.88 | 9.06 | 10.24 | 11.22 | 8.00 | **6.71** |
| A→C | 2.55 | 2.30 | 2.17 | 2.01 | 2.09 | **1.87** |
| B→C | 2.57 | 3.40 | 2.53 | **2.17** | 2.79 | 2.21 |
| D→C | 2.12 | 2.78 | 2.10 | 2.03 | 1.86 | **1.72** |
| A→D | 11.71 | 10.85 | **8.39** | 9.52 | 8.92 | 8.52 |
| B→D | 14.74 | 12.47 | 8.66 | 8.49 | **7.23** | 7.81 |
| C→D | 13.72 | 10.83 | 11.10 | 10.92 | 7.98 | **6.94** |
| Average | 12.23 | 8.37 | 8.35 | 7.76 | 7.63 | **6.34** |

**Table 5.** MAPE Performance. The best performance of each task is highlighted in bold.

| Task | FT | WDGRL | DAN | DANN-LSTM | DCORAL | Ours |
|------|------|-------|-------|-----------|--------|--------|
| B→A | 12.22 | 7.12 | 5.88 | 5.65 | 6.56 | **5.29** |
| C→A | 13.84 | 6.76 | 6.10 | 5.67 | 7.96 | **5.55** |
| D→A | 12.31 | 5.91 | 8.78 | 6.67 | 8.73 | **5.33** |
| A→B | 11.87 | 8.33 | 11.33 | 10.72 | 8.96 | **6.98** |
| C→B | 13.83 | 7.41 | 9.45 | 7.07 | 7.28 | **6.08** |
| D→B | 16.88 | 10.06 | 10.72 | 13.31 | 9.73 | **7.78** |
| A→C | 16.47 | 13.91 | 15.25 | 14.09 | 14.05 | **11.57** |
| B→C | 15.65 | 22.12 | 17.38 | 13.98 | 14.57 | **12.83** |
| D→C | 13.87 | 17.63 | 13.63 | 15.94 | 13.54 | **11.89** |
| A→D | 11.64 | 10.53 | **8.25** | 9.45 | 8.58 | 8.32 |
| B→D | 14.20 | 11.36 | 8.74 | 8.41 | **6.80** | 7.45 |
| C→D | 14.11 | 10.32 | 11.23 | 11.02 | 7.31 | **6.60** |
| Average | 13.91 | 10.95 | 10.56 | 10.17 | 9.50 | **7.97** |

In the domain adaptation tasks of the same target domain but different source domains, such as B→A, C→A, and D→A, the prediction error fluctuation of the method due to the change of the source domain is the slightest, which proves the transferability quantification based on information entropy success selectively transfers the knowledge in the source domain and mitigates negative effects where the low-correlation samples in the source domain lead to negative transfer.

The difference between the proposed method and other adversarial domain adaptation methods (DANN-LSTM and WDGRL) is the addition of the initial state fusion module to maintain the adversarial equilibrium. The proposed method has advantages in multiple tasks, and reduces RMSE by 1.57, MAE by 1.42, and MAPE by 2.2%; the adversarial equilibrium strategy based on initial state fusion effectively alleviates the adversarial disequilibrium problem. Domain structure information is supplemented in the intermediate features, which increases the reliability of domain discrimination. The domain discriminator supervises the feature extractor to achieve feature distribution alignment more effectively, thereby improving prediction accuracy.

The power load forecasting curves of the proposed method for one week from 0:00 on 14 March 2016, to 0:00 on 21 March 2016, are shown in Figure 6. The fitting degree between the prediction and the true value is high. The proposed method improves the load prediction accuracy effectively. However, the prediction error of the method for local peaks and valleys in the four fields is relatively large, and the power load mutation in field C is the most frequent, which means the user's personalized behavior is the most significant; thus, the prediction error of peaks is the largest, indicating that the prediction is easily affected by user personalized behavior. The transfer is not precise enough. Therefore, it is necessary to enhance the method's ability to learn domain-specific features, achieve more detailed selective transfer, suppress the occurrence of negative transfer more effectively, and further improve the prediction accuracy.

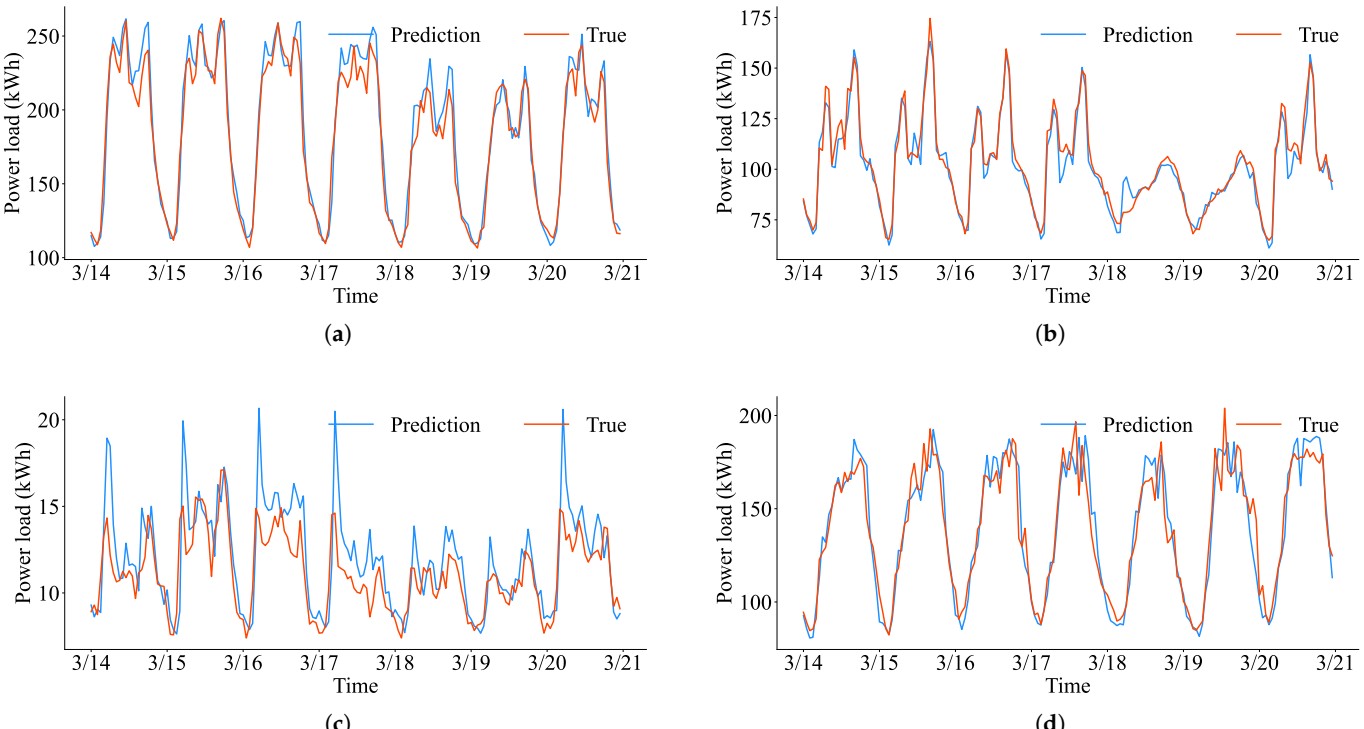

**Figure 6.** The power load forecasting curves for four buildings. (**a**) Task B→A; (**b**) task C→B; (**c**) task A→C; (**d**) task C→D.

Feature visualization is an important tool to measure the alignment degree of feature distribution. T-SNE [54] is widely used to visualize the high-dimensional data distribution in domain adaptation. The feature visualization results are shown in Figure 7. Red points correspond to the source domain, while blue ones correspond to the target domain. The more similar the source and target domain features are, the more effective the method is. In the proposed method, the source domain and target domain features have the smallest deviation, and the overlap between the two has a large proportion. Upon further analysis, it can be found that the features extracted and aligned by the proposed method are clustered,

and the boundaries of each cluster are sharper than the baseline method. Clusters represent the features that the method extracts from different aspects, it indicates that the initial state fusion strategy improves the domain discrimination ability of the domain discriminator, further supervising the feature extraction to extract domain-invariant features effectively during the adversarial training. There are few features that the proposed method fails to align relative to the baseline method, indicating that the proposed method effectively suppresses the low-correlation information in the source domain, and retains information that can be transferred to the target effectively.

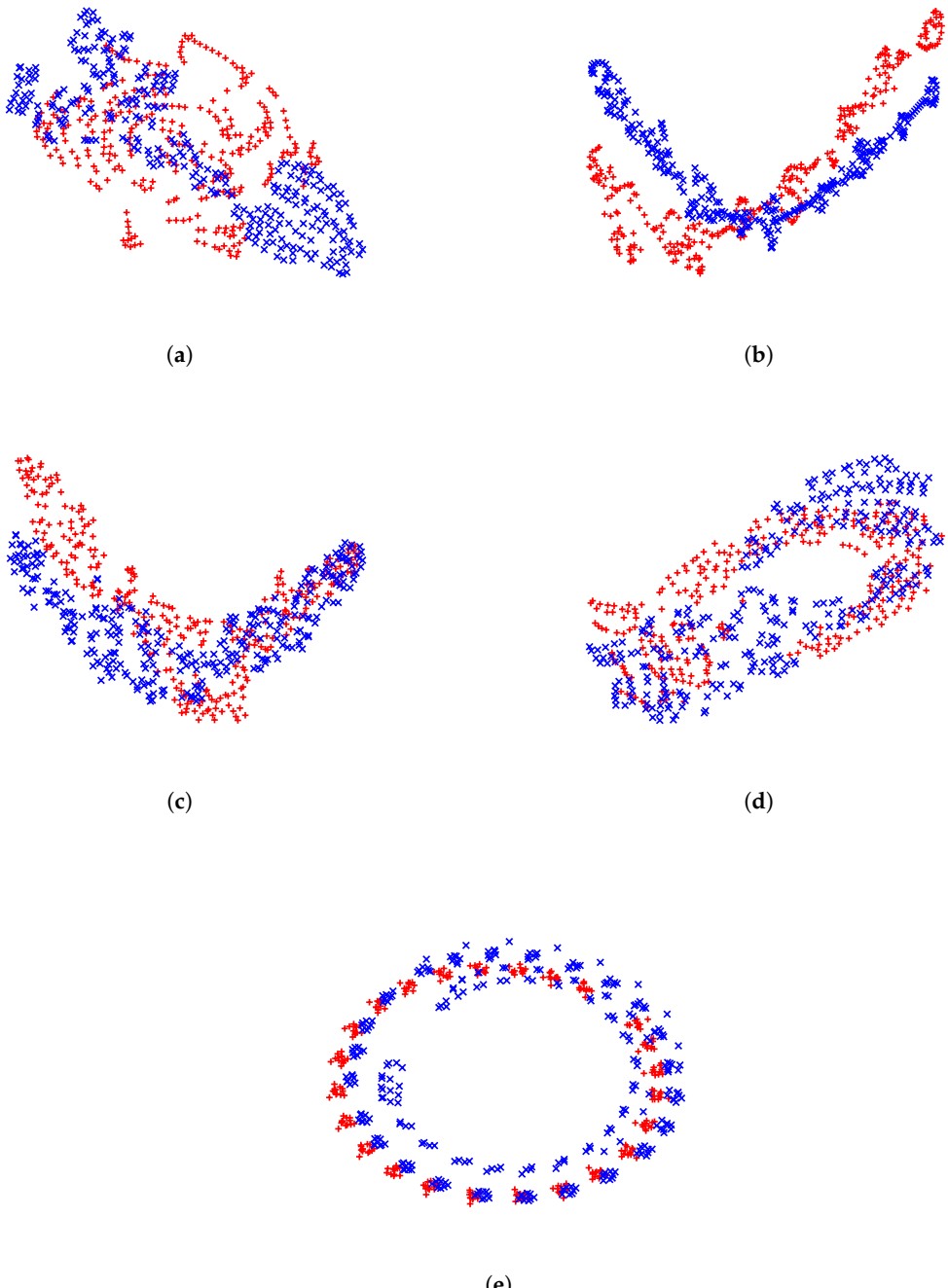

**Figure 7.** Feature visualization for different methods. Red points correspond to the source domain, while blue ones correspond to the target domain. (**a**) WDGRL; (**b**) DAN; (**c**) DANN-LSTM; (**d**) DCORAL; (**e**) Ours.

## 5. Conclusions

This paper focuses on the adversarial domain adaptation method and its application in power load forecasting. Domain adaptation alleviates the problem where traditional machine learning methods are limited by the amount of labeled data and the condition of IID; this has a strong significance for promoting intelligent power load forecasting systems. The adversarial domain adaptation method faces the problems of adversarial disequilibrium and a lack of transferability quantitation. This paper proposes corresponding solutions to the above two problems and conducts sufficient experimental verifications. The experimental results in the BDGP2 dataset prove that the proposed method gains a high power load prediction accuracy. This paper provides a research reference for solving the problems of adversarial disequilibrium and a lack of transferability quantitation, and provides an application reference for implementing power load forecasting based on the adversarial domain adaptation method. Furthermore, due to the strong personalization of users' electricity consumption behavior, the method does not perform well in the local peaks and valleys. Therefore, it is necessary to enhance the ability of the method to learn domain-specific features to achieve more refined selective transfer. Our future work will explore how to suppress the negative transfer better, and improve the prediction accuracy more effectively.

**Author Contributions:** Conceptualization, M.H. and J.Y.; methodology, M.H. and J.Y.; software, J.Y.; validation, M.H. and J.Y.; writing—original draft preparation, J.Y.; writing—review and editing, M.H.; funding acquisition, M.H. All authors have read and agreed to the published version of the manuscript.

**Funding:** This research was funded by two Guangdong Natural Science Foundation Projects (Grant No. 2021A1515011496 and Grant No. 2022A1515011370).

**Institutional Review Board Statement:** Not applicable.

**Informed Consent Statement:** Not applicable.

**Data Availability Statement:** Not applicable.

**Conflicts of Interest:** The authors declare no conflict of interest.

## Abbreviations

The following abbreviations are used in this manuscript:

| | |
|---|---|
| GANs | Generative Adversarial Networks |
| BDGP2 | Building Data Genome Project 2 |
| IID | Independent and Identical Distributions |
| MMD | Maximum Mean Discrepancy |
| MK-MMD | Multi-Kernel Maximum Mean Discrepancy |
| KL | Kullback–Leibler |
| JS | Jensen–Shannon |
| CORAL | CORrelation ALignment |
| 1DCNN | One-dimensional convolutional neural network |
| BiLSTM | Bidirectional Long Short Term Memory networks |
| ADF | Augmented Dickey Fuller |
| RMSE | Root Mean Square Error |
| MAE | Mean Absolute Error |
| MAPE | Mean Absolute Percentage Error |
| FT | FineTune |
| WDGRL | Wasserstein Distance Guided Representation Learning |
| DAN | Deep Adaptation Networks |
| DANN-LSTM | Domain Adversarial Neural Networks-Long Short Term Memory Networks |
| DCORAL | Deep CORAL |

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
