# Peer review of "Research on Adversarial Domain Adaptation Method and Its Application in Power Load Forecasting"

_mathematics, doi:10.3390/math10183223_

Round 1

Reviewer 1 Report

The English language needs serious revision. For example the sentence beginning at 21 and ending at 23 has no meaning. The same applies for the sentence at 254. Please read the manuscript carefully, before submitting ! It does not seem to have been read before submission!!!!

At the same time, in Table 1, explain the meaning of each figure in the column "Configuration". In Table 3, explain each of the acronyms and what each of these configurations does (FT, WDGRL and so on). A nomenclature section is a good idea (in which you can explain all acronyms).

In Fig 7 is not clear why your method is superior to all others when it comes to feature visualization. Please explain it thoroughly !

Reviewer 2 Report

Authors have proposed a method in which information entropy is employed to compute domain similarity into an adversarial domain adaptation method. This approach focuses on the problem of adversarial disequilibrium and lack of transferability quantitation. They have applied their methodology for power load forecasting using a public database, and have shown that results are consistently better than benchmark models.   In general, the paper is clear and well written. The use of the English language is appropriate. Please, consider the following comments:
  • Check grammar errors in Lines 254 and 260.
  • Line 265: I think authors want to say that subscript t refers to the "target" domain.
  • Results description can be improved in order to be able to replicate results.
  • What were the criteria employed for selecting the mentioned buildings data? Are these time series autocorrelated, seasonal, etc.? please provide additional insights about the data employed in experiments.
  • Exogenous variables are listed to some extent. Additional descriptions of how the experiments are necessary.

Reviewer 3 Report

The article presents a new method of short-term load forecasting. The idea of this method and experimental research of this method are presented. The idea of the method should be presented more precisely. It is presented too generally at present. The mathematical apparatus should be used to a greater extent. In the article, the authors state that the properties of the method, in particular the accuracy of the method, are to be demonstrated by the described experiment. However, this experiment is also presented too general. It is desirable that the data used be more accurately characterized. Why were such data taken into account? Are they representative? The answers to these questions are important because, on the basis of the results of the presented calculations, conclusions are drawn as to the properties of the proposed method.

Other comments:
1. The methods with which the proposed method is compared should be briefly characterized.

2. The text of the article should correspond with the figures.

3. All variables used should be described.

4. All acronyms should be explained before their first use. 

5. English should be improved.

Round 2

Reviewer 1 Report

Please re-examine the topic of your sentences. For example, rows 328-329. 

Reviewer 3 Report

I have no further comments.

Author Response

Dear Reviewer:

We sincerely thank you for your enthusiastic work.

Thanks again for your valuable comments and suggestions.

Sincerely,
Min HUANG & Jinghan YIN